# Model-enhanced Vector Index

**Hailin Zhang**[1,2,*] **Yujing Wang**[2,†] **Qi Chen**[2] **Ruiheng Chang**[2] **Ting Zhang**[2]
**Ziming Miao**[2] **Yingyan Hou**[2,3,*] **Yang Ding**[2,4,*] **Xupeng Miao**[5] **Haonan Wang**[6]
**Bochen Pang**[2] **Yuefeng Zhan**[2] **Hao Sun**[2] **Weiwei Deng**[2] **Qi Zhang**[2]
**Fan Yang**[2] **Xing Xie**[2] **Mao Yang**[2] **Bin Cui**[1,7,†]
[1]School of Computer Science & Key Lab of High Confidence
Software Technologies, Peking University    [2]Microsoft
[3]Aerospace Information Research Institute & Key Laboratory of Target
Cognition and Application Technology, Chinese Academy of Sciences
[4]Institute of Information Engineering, Chinese Academy of Sciences
[5]Carnegie Mellon University   [6]National University of Singapore
[7]National Engineering Laboratory for Big Data Analysis and Applications, Peking University
[1]{z.hl, bin.cui}@pku.edu.cn
[2]{yujwang, cheqi, ruihengchang, tinzhan, zimiao, bopa}@microsoft.com
[2]{yuefzh, hasun, dedeng, qizhang, fanyang, xingx, maoyang}@microsoft.com
[3]houyy@aircas.ac.cn  [4]dingyang@iie.ac.cn
[5]xupeng@cmu.edu  [6]haonan.wang@u.nus.edu

## Abstract

Embedding-based retrieval methods construct vector indices to search for document representations that are most similar to the query representations. They are widely used in document retrieval due to low latency and decent recall performance. Recent research indicates that deep retrieval solutions offer better model quality, but are hindered by unacceptable serving latency and the inability to support document updates. In this paper, we aim to enhance the vector index with end-to-end deep generative models, leveraging the differentiable advantages of deep retrieval models while maintaining desirable serving efficiency. We propose Model-enhanced Vector Index (MEVI), a differentiable model-enhanced index empowered by a twin-tower representation model. MEVI leverages a Residual Quantization (RQ) codebook to bridge the sequence-to-sequence deep retrieval and embedding-based models. To substantially reduce the inference time, instead of decoding the unique document ids in long sequential steps, we first generate some semantic virtual cluster ids of candidate documents in a small number of steps, and then leverage the well-adapted embedding vectors to further perform a fine-grained search for the relevant documents in the candidate virtual clusters. We empirically show that our model achieves better performance on the commonly used academic benchmarks MSMARCO Passage and Natural Questions, with comparable serving latency to dense retrieval solutions.

## 1 Introduction

A web search engine basically consists of two key stages, document retrieval and ranking [52, 24, 47, 9]. The document retrieval stage retrieves candidate documents related to the query, while the ranking stage re-ranks the documents using precise ranking scores. Accurate ranking models are usually deep neural networks that predict query and document relevance scores [17, 13]. To guarantee low latency

---

*The work was done at Microsoft.

†Corresponding authors.

37th Conference on Neural Information Processing Systems (NeurIPS 2023).

in online systems, such ranking models can only support hundreds or thousands of documents per query, requiring the retrieval stage to recall relevant documents precisely. Therefore, improving the quality of retrieval models is of great significance to improve the effectiveness of web search engines.

Existing methods of document retrieval can be roughly divided into three categories, namely term-based, embedding-based and generation-based methods [14, 46]. Term-based methods, also called sparse retrieval, build an inverted index upon words or phrases from the whole corpus for searching [4, 56]. However, they are not aware of document semantics information and may miss target documents with different wordings. Embedding-based methods, also called dense retrieval, are proposed to leverage semantics information [52, 28]. They encode queries and documents to dense embedding vectors by a twin-tower architecture, then build a vector index and apply Approximate Nearest Neighbor (ANN) search to retrieve relevant documents for queries [12]. Such methods are widely used in real applications, since they have decent recall performance and low serving latency. Although these methods prevail in industry, they cannot fully leverage the power of deep neural networks because they are not end-to-end differentiable due to the necessity of ANN index. They suffer from the discrepancy between brute-force K-Nearest Neighbor (KNN) and ANN. The discrepancy leads to a non-trivial gap in recall performance of KNN and ANN [49].

Recently, generation-based methods [44, 46] are proposed to tackle the limitation of embedding-based retrieval solutions. Typically, a generation-based model adopts a sequence-to-sequence architecture, generating document identifiers directly based on the given queries. To achieve better results, the document identifiers need to reflect effective priors of the document semantics. For example, SEAL [2] leverages the n-grams itself as identifiers. In DSI [44] and NCI [46], documents are organized as a tree by hierarchical KMeans clustering [15], and identifiers are the codes of paths from the root to the leaf nodes. Though such methods reach better recall performance than embedding-based retrieval models on small corpus (containing less than 1 million documents), they have difficulty in scaling to larger corpus (containing more than 10 million documents), and can not be served online in industrial systems due to high latency and immutable corpus. To train a deep retrieval model that indexes larger dataset, we need much more parameters to memorize all documents semantics, and the decoder needs to compute more times by beam search at inference. Another blocking issue is that the tree index is fixed during training and inference, posing a significant challenge for adding or removing documents. As a result, the model needs to be retrained every time the corpus changes. Overall, generation-based retrieval methods are time-consuming and only applicable to static small corpus. To the best of our knowledge, they have not been successfully used in real applications with large-scale corpus due to these non-trivial challenges.

In this paper, we aim to address the aforementioned constraints of model-based methods while preserving their benefits. We propose Model-enhanced Vector Index (MEVI), which enjoys high recall performance as well as fast retrieval speed on large-scale corpus. Specifically, we construct a Residual Quantization (RQ) codebook to cluster the documents first. The RQ codebook preserves the hierarchical structure of the document clusters, which is inherently suitable for autoregressive generation. Thereafter, we build a sequence-to-sequence model to encode user queries and directly generate virtual cluster identifiers according to the RQ codebook; then we conduct efficient ANN method to search embedding vectors which are semantically relevant to the virtual cluster. During training, ground-truth data and augmented query-document pairs are fed into the model. During inference, the top K document clusters are retrieved from the RQ codebook via beam search on the decoder, and an ANN search process is conducted based on the query embedding as well as the retrieved document embeddings corresponding to the clusters. Our MEVI design solves the limitations of both traditional embedding-based methods and generation-based methods. On the one hand, we can limit the RQ codebook to a moderate size to reduce the computation time of the autoregressive decoder and thus ensure low latency. On the other hand, MEVI enables insertion or removal of documents to RQ-based index structures, such that new documents can be recalled if their virtual clusters with related semantics can be generated by the sequence-to-sequence model. By selecting a suitable size for the RQ codebook, we can balance the recall performance and the inference latency. This allows us to leverage the high efficiency of ANN and the accurate recall of deep retrieval models simultaneously. The code of MEVI is available at: https://github.com/HugoZHL/MEVI.

Our **contributions** are highlighted as follows.

- For the first time, we demonstrate that a novel-designed generation-enhanced model is able to handle a large corpus with millions of documents, reaching high recall performance and low serving

latency at the same time. The proposed MEVI provides a unified solution for document retrieval, incorporating the advantages of both embedding-based methods and generation-based methods.

- In our experiments, MEVI significantly improves the recall performance compared to existing methods, achieving **+3.62% / +7.32% / +10.54%** of MRR@10 / R@50 / R@1000 on the MS-MARCO Passage dataset and **+5.04% / +5.46% / +5.96%** of R@5 / R@20 / R@100 on the Natural Questions dataset with low latency capable of serving. Importantly, We also validate MEVI's capability in addressing dynamic corpora effectively.

- We propose a novel RQ-based sequence-to-sequence model to support training and serving on large-scale retrieval datasets. As verified by experiments and analysis, the RQ structure allows for dynamic updates of documents and enables better model performance with low serving latency.

## 2 Related work

Document retrieval aims to find relevant documents that match a user's query. The core idea of document retrieval is to build an index for a vast number of documents to guarantee fast query response. Document retrieval methods can be divided into three categories.

Traditional sparse retrieval methods build on top of an inverted index using term matching metrics such as TF-IDF [41], query likelihood [23], and a hard-to-beat baseline BM25 [42] in industry-scale web search. Unfortunately, these methods do not take into account the semantics of documents, which can lead to the overlooking of target documents with different wordings.

Dense retrieval methods represent queries and documents using dense embedding vectors, and build ANN index to speed up the search. These methods take advantage of recent advances in pre-trained language models to encode queries and documents into dense representations, such as BERT [8] and RoBERTa [27] . During training, dense retrieval generally follows the contrastive learning paradigm. During inference, ANN methods are applied to search relevant documents, such as tree [1, 25], locality sensitive hashing [6], neighbor graph index (e.g., HNSW [30], DiskANN [43], HMANN [40]), the combination of graph index and inverted index (e.g., SPANN [3]). Sparse retrieval and dense retrieval can be combined [29] to achieve better search quality. However, the discrepancy between KNN and ANN presents a challenge for these methods, particularly in scenarios where there is a significant distribution difference between documents and queries. Besides document retrieval, embeddings are also utilized in cross-modal retrieval [55, 18], graph [5, 19], recommendation [32, 33], etc.

Autoregressive retrieval is a new paradigm to retrieve relevant documents. It uses an end-to-end autoregressive model that takes queries as input and directly generates document identifiers as output. Differentiable Search Index (DSI) [44] and Neural Corpus Indexer (NCI) [46] are the first attempts to propose differentiable indexes for semantics search that directly map queries to relevant document identifiers. DSI generates document identifiers with document tokens, and designs multi-task training for augmentation. NCI builds a tree index by hierarchical KMeans clustering, and represents each file by the path from root to leaf on the tree. It designs a weight-adaptive decoder and uses augmented data to further improve performance. SEAL [2] leverages autoregressive models in another way, generating n-grams and retrieving documents through a pre-built FM-index for n-grams. However, they are all limited by unacceptable serving latency and the inability to handle document updates.

## 3 Model-enhanced vector index

In this paper, we propose a novel Model-enhanced Vector Index (MEVI) to address the above limitations of both embedding-based dense retrieval methods and generation-based autoregressive retrieval methods. The goal of MEVI is to achieve both fast retrieval speed, like dense retrieval methods, and high recall performance, like autoregressive retrieval methods, by leveraging the strengths of both approaches.

MEVI adopts an autoregressive sequence-to-sequence model as the enhancing model and a well-adapted twin-tower representation model as the vector index. The twin-tower model encodes queries and documents into embedding vectors for common embedding-based retrieval. The autoregressive sequence-to-sequence model takes the same queries as input and outputs the most relevant document clusters. To organize documents into hierarchical clusters, MEVI employs Residual Quantization [31], which first clusters the document embeddings and then iteratively clusters the residual embeddings. During each clustering pass, the residual embeddings are computed by subtracting the cluster centroid embeddings from the embeddings used in the previous pass. RQ is essentially a hierarchical

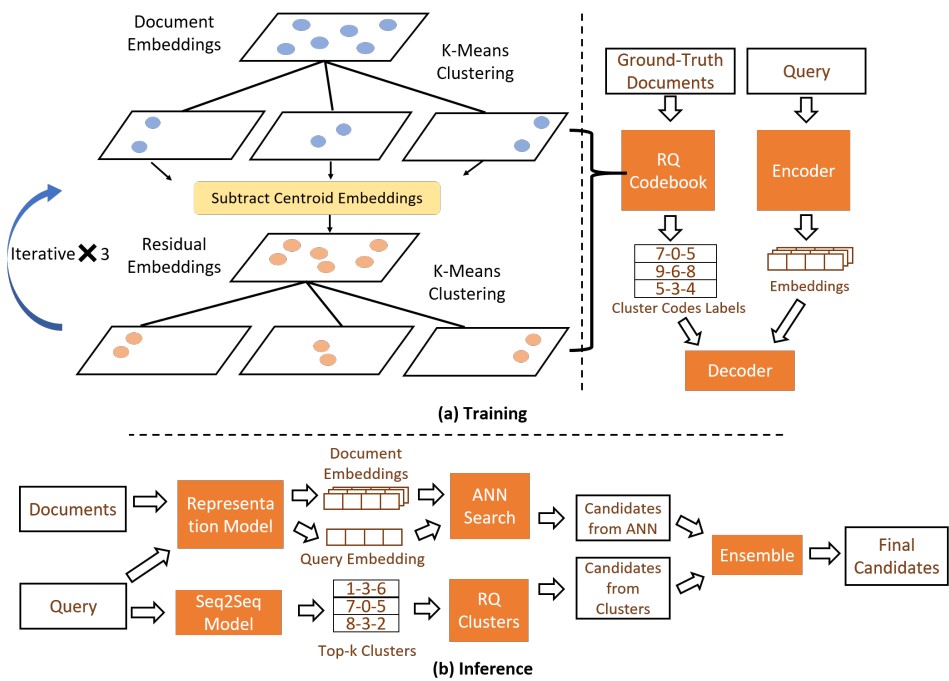

Figure 1: Overview of Model-enhanced Vector Index (MEVI). (a) Training. Document embeddings and residual embeddings are iteratively clustered to form the RQ structure; then the RQ structure provides cluster codes as labels for the training of sequence-to-sequence model. (b) Inference. The query is fed into the representation model and the sequence-to-sequence model respectively to get candidates from ANN and clusters; then ensemble strategies are performed to get final candidates.

KMeans [15] algorithm, where similar documents are assigned to the same or close clusters. Document clustering not only works in traditional clustering-based retrieval [45, 26], but is also suitable for autoregressive sequence-to-sequence models as mentioned in DSI [44] and NCI [46]. Once the results are obtained from the autoregressive model and the twin-tower model, MEVI ensembles the results to form the final document candidates. The whole workflow is shown in Figure 1.

### 3.1 Organizing documents with Residual Quantization

Twin-tower representation models [38, 11, 51] are designed for dense retrieval, also known as embedding-based retrieval. They separately encode queries and documents into embeddings, and perform similarity search (e.g. Euclidean, cosine, or inner product) on the embeddings to retrieve relevant documents for given queries. Our method utilizes information from autoregressive sequence-to-sequence models to significantly and accurately narrow down the range of candidate sets. We use the state-of-the-art twin-tower models AR2 [54] and T5-ANCE [51] in our experiments. It is worth noting that the improvement can theoretically be applied to any twin-tower representation models.

To build an autoregressive sequence-to-sequence task, documents need to be organized into some kind of data structures. In DSI [44] and NSI [46], documents are hierarchically clustered through the KMeans algorithm [15] to form a tree, and the T5 model [39] is used to encode the query and decode into a document identifier represented by a path on the tree. The tree structure, however, cannot support large-scale corpus in real applications due to its long sequential decoder steps.

Vector quantization is a useful method to compress document embedding representations. Instead of the commonly used Product Quantization which splits embeddings along dimension, we leverage Residual Quantiztaion (RQ) to preserve the hierarchical semantics of documents. The construction of RQ codebook is a process of iterative clustering. Concretely, assume the initial document embedding is $X \in \mathbb{R}^{n \times d}$, the first KMeans clustering is performed on $X$ to get the centroid embeddings $B_1 \in \mathbb{R}^{b \times d}$, also known as codewords, and the one-hot clustering assignment $C_1 \in \{0,1\}^{n \times b}$, also known as codes. Then the second clustering is performed on the residual embeddings $R_1 = X - C_1 B_1$,

rather than the original document embeddings $X$. Codewords $B_2$ and codes $C_2$ are obtained in this step. Then the residual $R_2 = R_1 - C_2 B_2$ is fed into the third clustering, and the process continues till a pre-defined number of steps $m$ is reached. After all the clustering process, the overall codebook is obtained as $B = [B_1, B_2, ..., B_m] \in \mathbb{R}^{mb \times d}$, and the overall codes for documents are $C = [C_1^T, C_2^T, ..., C_m^T]^T \in \{0, 1\}^{n \times mb}$; the iterative clustering can be seen as a solution to the minimization problem $\min ||X - CB||_2^2$. The explanation of symbols are detailed in Table 1.

Table 1: Notations.

| Notation | Explanation |
|---|---|
| $n$ | The number of documents |
| $d$ | Embedding dimension |
| $\log_2 b$ | The bit number of compact codes |
| $b$ | The number of centroid embeddings for each clustering |
| $m$ | The number of clustering layers, i.e. the length of cluster-id |
| $X \in \mathbb{R}^{n \times d}$ | All document embedding |
| $B_i \in \mathbb{R}^{b \times d}, i = 1, ..., m$ | Centroid embeddings of clusters |
| $B \in \mathbb{R}^{mb \times d}$ | Overall centroid embeddings |
| $C_i \in \{0, 1\}^{n \times b}, i = 1, ..., m$ | Code-style assignment of documents |
| $C \in \{0, 1\}^{n \times mb}$ | Overall cluster assignment |
| $\widetilde{C} \in \{0, 1, 2, ..., b-1\}^{n \times m}$ | Overall compact cluster assignment |
| $k$ | The number of retrieved clusters |
| $K$ | The number of retrieved documents (for R@K, MRR@K) |
| $c_i, i = 1, ..., k$ | Top-k retrieved clusters |
| $R_i \in \mathbb{R}^{n \times d}, i = 1, ..., m-1$ | Residual embeddings after clustering |
| $s_0, s_c$ | Scoring function for embedding similarity and cluster ranking |
| $r_c$ | Cluster ranking function |
| $\alpha, \beta$ | Hyperparameters of the ensemble process |

Both RQ codebooks and hierarchical trees are constructed using iterative clustering. The difference is that RQ clusters the residuals and maintains a fixed step length. Considering that the clustering process is essentially the classification of documents, RQ's clustering of residuals can be regarded as classifying documents in a more and more refined manner. On the other hand, all codes in RQ are of length $m$, making the latency of the autoregressive process stable and acceptable.

After the RQ construction process, we obtain the codes of documents $C$, or a compact version $\widetilde{C} \in \{0, 1, 2, ..., b-1\}^{n \times m}$ instead of one-hot vectors, with each code an index of the codebook in the corresponding step. In the decoding process of autoregressive sequence-to-sequence models, the code $\widetilde{C}$ is generated step by step during inference until step $m$, with the vocabulary of each step $\{0, 1, 2, ..., b-1\}$. We consider documents to be from the same cluster only if their codes are identical. Therefore, we regard the results from the sequence-to-sequence model as clusters with top-k probability of containing relevant documents. With the mapping of document to code $\widetilde{C}$, the sequence-to-sequence model can be trained with a large collection of $< query, code >$ pairs derived from $< query, document >$ pairs.

### 3.2 Ensemble process

Given the top-k retrieved clusters $\{c_1, c_2, ..., c_k\}$ from the autoregressive sequence-to-sequence model, we incorporate such information into embedding-based retrieval. The rest of this section explores some possible options to ensemble the top-k cluster information.

**Search only in candidates from top-k clusters.** A straight-forward strategy is to only search relevant embeddings within top-k clusters. This strategy assumes that the sequence-to-sequence model has already retrieved relevant documents by retrieving top-k related clusters, greatly reducing the number of candidates for embedding similarity search.

**Ensemble the original embedding search results with the candidates from the top-k clusters.** Considering that the RQ structure may be inaccurate, and the beam search process is a lossy greedy algorithm, the retrieved top-k clusters may not contain all the relevant documents. A more robust

method is to combine candidates from common embedding-based retrieval and top-k clusters. The original similarity score is not applicable for re-ranking these candidates, since the first candidate set already contains the best documents under this similarity metric. Therefore, we design a new score for re-ranking, also leveraging the information of retrieved top-k clusters. Assume $s_0$ denotes the original scoring function for embedding similarity, $s_c$ denotes the cluster scoring function of documents, $\alpha$ denotes the coefficient of cluster scores, $x$ is a random document, the new score is:

$$s(x) = s_0(x) + \alpha \cdot s_c(x)$$

The cluster scoring function can be a function of the cluster ranking:

$$s_c(x) = \frac{1}{\beta \cdot r_c(x) + 1}$$

where $r_c$ denotes the cluster ranking function of documents, $\beta$ is another hyperparameter. For documents that are not in the top-k clusters, we simply use zero or a value smaller than the minimum cluster score.

## 4 Experiments

In this section, we conduct various experiments to verify the effectiveness and the efficiency of MEVI. We first introduce the datasets and the evaluation metrics to be used (Section 4.1), and then describe the implementation details of MEVI and other baseline methods (Section 4.2). We demonstrate and analyze the effectiveness of MEVI on two large-scale datasets (Section 4.3, 4.4), examine the ability of MEVI to handle unseen documents (Section 4.5), and explore the effectiveness of RQ (Section 4.6). We also examine the efficiency of MEVI (Section 4.7), study the impact of hyperparameters (Section 4.8) and apply our method to practical applications (Appendix A).

### 4.1 Datasets & metrics

**Datasets.** We conduct experiments on two widely used large-scale document retrieval benchmarks, i.e. MSMARCO [34] Passage Retrieval dataset and Natural Questions (NQ) [22] dataset. MSMARCO is a question answering dataset formulated by Microsoft in 2016. It consists of real Bing questions and human-generated answers. The version we use is the MSMARCO Passage Retrieval dataset, which contains 8.8 million passages and around 510,000 queries. NQ is a large open-domain question answering dataset containing queries collected from Google search logs. DPR [21] selected around 62,000 factoid questions and processed all the Wikipedia articles (21 million) as the collection of passages. We use the DPR version for evaluation. For both datasets, we use their predetermined training and validation splits for evaluation.

**Metrics.** To reflect whether relevant documents are retrieved given a user query, we employ Recall@K and MRR@K, two widely accepted information retrieval evaluation metrics. R@K (short for Recall@K) measures the proportion of relevant items found among the retrieved top-K candidates. A higher recall means that there are more relevant documents among the top-K candidates. MRR@K (mean reciprocal rank) calculates the reciprocal of the rank at which the first relevant document is retrieved within top-K candidates. A higher MRR indicates that the relevant document is retrieved with a higher rank.

### 4.2 Implementation details

**RQ codebook.** By default, we use an RQ codebook with 4 clustering layers and 5 bits per layer. In each layer, we use the default KMeans algorithm provided by scikit-learn [37], where the number of centroids is $b = 2^5 = 32$. The entire RQ has $32^4 = 1,048,576$ clusters, and each cluster contains an average of 8 and 20 documents in two datasets. We tested other configurations of RQ in Section 4.6.

**Sequence-to-sequence model.** We leverage Neural Corpus Indexer (NCI) [46], the state-of-the-art sequence-to-sequence model for document retrieval. We replace its original decoder vocabularies with our RQ structure, and train NCI upon the processed $<query, code>$ pairs. For data augmentation, we employ the "document as query" technique and use DocT5Query [35] as suggested in NCI, generating 10 queries per document for MSMARCO Passage, and 1 query per document for NQ. We use the same optimizer and learning rate as NCI; however, we use a larger batch size, 256 per GPU, and disable Rdrop to reduce the overall training cost. For inference, we use beam search with a beam

size of 10 to 1000. All experiments of NCI are conducted on an NVIDIA V100 GPU cluster with 32GB memory per GPU, and each job runs on 8 GPUs.

**Twin-tower representation model.** We use T5-ANCE [36] to generate embeddings for RQ clustering. T5-ANCE is an effective embedding-based retrieval model which replaces the backbones of ANCE [51] with T5 [39]. We use T5-ANCE and AR2 [54] to generate embeddings for ensembling. AR2 is the state-of-the-art twin tower retrieval model. These two methods have different backbone models, indicating that our method can be applied to different twin-tower representation models. To obtain the search result from embedding-based retrieval, we adopt HNSW [30], a novel ANN search technique implemented in Faiss [20], with 256 neighbors per vertex. We have also tried more neighbors per vertex but got no gain compared to 256.

**Ensemble.** All the ensemble strategies require searching among the top-k clusters retrieved by the sequence-to-sequence model. Instead of performing ANN search as in traditional embedding-based retrieval, we employ brute-force search in top-k clusters. This is because the number of candidates in the top-k clusters is relatively small, typically on the order of ten thousand, given a beam size of 1000. The hyperparameters of the ensemble are determined through grid-search. We study the impact of hyperparameters in Section 4.8.

**Baselines.** We compare our method with sparse retrieval methods BM25 [42] and SPLADE [10], dense retrieval methods T5-ANCE and AR2 with ANN, and model-based retrieval NCI. The comparisons with T5-ANCE/AR2 and NCI also reflect the gain of ensemble strategies. For NCI, we build a hierarchical tree with a depth of 8. The other hyperparameters of the baseline methods remain the same as the original papers.

## 4.3 Results on MSMARCO Passage

Table 2 presents the retrieval results on MSMARCO Passage dataset. We compare different ensemble strategies of MEVI with baselines. We report the results of searches in the top-10, top-100, and top-1000 clusters, and the ensemble of these results with the embedding-based retrieval methods. Among all the tested methods, the ensemble of embedding-based retrieval and generation-based retrieval outperforms all baselines. Compared to the ANN baseline, MEVI improves 3.62% / 2.62% for MRR@10, 7.32% / 5.43% for Recall@50, 10.54% / 8.00% for Recall@1000 on AR2 / T5-ANCE respectively. The more clusters to be considered, the better the retrieval performance. Even without ensemble, vanilla MEVI performs better than dense retrieval methods when considering top-1000 clusters. After ensemble, MEVI can achieve better results than ANN with only top-10 clusters, which shows that the ensemble combines the advantages of ANN and top-k clusters, making them complement each other. Compared to NCI, MEVI performs better especially on MRR@10, because MEVI utilizes generation-based retrieval with smaller decoder steps, allowing the model to learn a better ranking.

Table 2: Experiment results on MSMARCO Passage (Dev).

| Method | MRR@10 | R@50 | R@1000 |
|---|---|---|---|
| BM25 | 18.7 | 59.2 | 85.7 |
| SPLADE | 32.2 | / | 95.5 |
| T5-ANCE (HNSW) | 33.21 | 77.30 | 88.61 |
| AR2 (HNSW) | 35.54 | 78.80 | 87.11 |
| NCI | 26.18 | 74.68 | 92.44 |
| MEVI Top-10 Clus | 32.05 | 63.25 | 66.82 |
| MEVI Top-100 Clus | 35.16 | 79.14 | 88.22 |
| MEVI Top-1000 Clus | 35.76 | 82.37 | 95.17 |
| MEVI Top-10 Clus & T5-ANCE (HNSW) | 35.22 | 81.29 | 93.21 |
| MEVI Top-100 Clus & T5-ANCE (HNSW) | 35.60 | 82.27 | 95.62 |
| MEVI Top-1000 Clus & T5-ANCE (HNSW) | **35.83** | **82.73** | **96.61** |
| MEVI Top-10 Clus & AR2 (HNSW) | 37.00 | 82.64 | 93.46 |
| MEVI Top-100 Clus & AR2 (HNSW) | 38.42 | 84.52 | 96.23 |
| MEVI Top-1000 Clus & AR2 (HNSW) | **39.16** | **86.12** | **97.65** |

## 4.4 Results on Natural Questions

Table 3 presents the retrieval results on Natural Questions dataset. Compared to the ANN baseline, MEVI improves 5.04% for Recall@5, 5.46% for Recall@20, 5.96% for Recall@100 on AR2 model. Vanilla MEVI also performs better than AR2 (HNSW) when searching among top-1000 clusters. After ensemble, MEVI can outperform AR2 (HNSW) with only 10 clusters.

Table 3: Experiment results on Natural Questions (Test).

| Method | R@5 | R@20 | R@100 |
|---|---|---|---|
| BM25 | / | 59.1 | 73.7 |
| AR2 (HNSW) | 70.89 | 78.50 | 83.02 |
| MEVI Top-10 Clus | 59.61 | 66.45 | 71.63 |
| MEVI Top-100 Clus | 70.33 | 77.23 | 81.77 |
| MEVI Top-1000 Clus | 75.57 | 82.83 | 87.31 |
| MEVI Top-10 Clus & AR2 (HNSW) | 74.10 | 82.11 | 86.43 |
| MEVI Top-100 Clus & AR2 (HNSW) | 74.43 | 82.71 | 87.51 |
| MEVI Top-1000 Clus & AR2 (HNSW) | **75.93** | **83.96** | **88.98** |

## 4.5 Dynamic update of documents

In order to test whether MEVI can support the dynamic update of documents, we randomly remove 10% of the documents in the MSMARCO Passage dataset during training, and add these 10% documents back during evaluation. We compare MEVI with the embedding-based retrieval method T5-ANCE. Since NCI only maintains a fixed document tree that does not support dynamic update of documents, we do not include NCI in this experiment.

Table 4 shows the results. Performance drop compared to normal training are listed in the parentheses. All tested methods lose some model accuracy when 10% of the documents are discarded during training. MEVI still performs well compared to ANN, showing the ability to handle unseen documents. From experiments we also find that the recall drops less as more clusters are retrieved, because more clusters provide more possible semantics information about unseen documents.

Table 4: Train with random 90% documents, and evaluate with all documents.

| Method | MRR@10 | R@50 | R@1000 |
|---|---|---|---|
| T5-ANCE (HNSW) | 29.34 (3.87) | 73.09 (4.21) | 86.26 (2.35) |
| MEVI Top-10 Clus | 28.20 (3.85) | 55.43 (7.82) | 58.81 (8.01) |
| MEVI Top-100 Clus | 33.48 (1.68) | 74.16 (4.98) | 82.68 (5.54) |
| MEVI Top-1000 Clus | 35.38 (0.38) | 81.06 (1.31) | 93.40 (1.77) |

## 4.6 Residual quantization

In this section, we first analyze the advantages of RQ over ordinary KMeans clustering, and then study the impact of RQ codebook configuration.

We choose RQ instead of normal hierarchical KMeans in MEVI. Compared to hierarchical KMeans that focuses on more fine-grained clustering within large clusters in each layer, RQ adopts residual information to address the errors of the previous layer, making the clustering results more precise and robust. We conduct an experiment on MSMARCO Passage to compare these two algorithms in MEVI. In the experiments, we set layer depth to 4 and the number of centroids per layer to 32. In Table 5, RQ generally achieves better recall than normal KMeans with the same number of clusters.

In Table 6, we use $RQ(x \times y)$ to represent an RQ codebook with $x$ layers, $y$ bits ($2^y$ centroids) per layer, and a total of $2^{xy}$ clusters. For simplicity, we train the models on MSMARCO Passage with only 1 generated query per document for augmentation. We report MRR@10, Recall@100, the total recall of top-k clusters, and the average number of retrieved documents per query. The numbers of

Table 5: Comparison of KMeans and RQ in MEVI.

| Method | MRR@10 | R@50 | R@1000 |
|---|---|---|---|
| MEVI-KMeans Top-10 Clus | 31.62 | 65.09 | 68.25 |
| MEVI-KMeans Top-100 Clus | 34.82 | 77.30 | 87.93 |
| MEVI-KMeans Top-1000 Clus | 35.65 | 81.01 | 94.17 |
| MEVI-RQ Top-10 Clus | 32.05 | 63.25 | 66.82 |
| MEVI-RQ Top-100 Clus | 35.16 | 79.14 | 88.22 |
| MEVI-RQ Top-1000 Clus | **35.76** | **82.37** | **95.17** |

layers and bits both determine the size of cluster-id space, while the number of retrieved clusters $k$ determines the number of retrieved documents per query. To fairly compare these configurations, we would like to align the number of retrieved documents per query to the same order of magnitude. From $RQ(3 \times 4)$ to $RQ(5 \times 5)$, the number of clusters increases, and the number of documents in each cluster decreases, so we enlarge the number of clusters from 3 to 1000 to align the number of the documents per query. In general, the larger the number of candidate documents, the more likely the model is to recall the correct document. With the same order of magnitude of retrieved documents, $RQ(4 \times 5)$ performs well with moderate number of retrieved clusters. Although $RQ(5 \times 5)$ performs better in this case, it needs to retrieve more clusters to maintain an appropriate number of document candidates, which poses serious memory challenges. Larger configurations also mean more decoder steps, resulting in larger inference latency. Hence for MSMARCO Passage dataset, $RQ(4 \times 5)$ is more proper. We leave the support for larger RQ in future work.

Table 6: Model performance with different RQ configurations.

| Configuration | # Clusters | MRR@10 | R@100 | R@Clusters | # Docs / Query |
|---|---|---|---|---|---|
| $RQ(3 \times 4)$ | 3 | 26.38 | 55.05 | 58.83 | 13318 |
| $RQ(4 \times 4)$ | 10 | 29.15 | 63.28 | 67.28 | 9750 |
| $RQ(4 \times 5)$ | 100 | 33.36 | 76.14 | 80.71 | 4972 |
| $RQ(5 \times 4)$ | 100 | 32.57 | 72.92 | 77.11 | 4764 |
| $RQ(5 \times 5)$ | 1000 | 34.12 | 79.69 | 85.62 | 3999 |

## 4.7 Efficiency

We test the online serving latency of MEVI on a single A6000 card. Since the inference time is affected by the number of retrieved clusters, we list the latency of different settings in Table 7. For MEVI and NCI, the latency mainly comes from the stepwise generation of the autoregressive sequence-to-sequence model. Latency increases with the number of clusters, because larger beam size leads to larger autoregressive generation time, as well as larger number of candidates to be searched. NCI has much longer decoder steps than MEVI, resulting in a significant increase in latency. Latency of ANN includes the query encoding time and the HNSW searching time; it is the fastest method, which is appealing to real applications. Compared to ANN, MEVI can achieve a good performance and latency trade-off when $k = 10$, improving the MRR@10 metric by more than 2 absolute points with a less than 100ms latency. This is acceptable online, as by adopting some acceleration techniques (e.g., FP16, ONNX Runtime [7]), we can further improve the average serving latency to less than 40ms.

Table 7: Trade-off between serving latency and model performance.

| Method | MRR@10 | Latency (ms) |
|---|---|---|
| T5-ANCE (HNSW) | 33.21 | 19.71 |
| NCI | 26.18 | 2899.17 |
| MEVI Top-10 & T5-ANCE (HNSW) | 35.22 | 96.87 |
| MEVI Top-100 & T5-ANCE (HNSW) | 35.60 | 222.55 |
| MEVI Top-1000 & T5-ANCE (HNSW) | 35.83 | 1662.84 |

## 4.8 Hyperparameters

We try different $K$ for Recall and MRR on MSMARCO Passage in Figure 2. "HNSW" means T5-ANCE with HNSW index, "Ensemble" means the ensemble result of T5-ANCE (HNSW) and MEVI Top-1000 Cluster. MEVI ensemble results consistently outperform T5-ANCE (HNSW).

In the ensemble process, we search the values of hyperparameters $\alpha$ and $\beta$ in a proper range and choose the configuration with the best model metrics. Since the ensemble process does not incur additional training and inference cost, the time for grid search can be ignored. Figure 3 shows the MRR@10 of MEVI Top-1000 & T5-ANCE (HNSW) on MSMARCO Passage with different $\alpha$ and $\beta$. The optimal configuration fully leverages the information of both components. Our ensemble method is also robust to hyperparameters, as the ensemble result outperforms the baseline (MEVI Top-1000 Clus 35.76) when $0.3 \leq \alpha \leq 0.7$ and $0.01 \leq \beta \leq 0.03$.

Figure 2: Recall and MRR at different K.

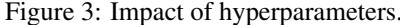

(a) Recall@K.           (b) MRR@K.

Figure 3: Impact of hyperparameters.

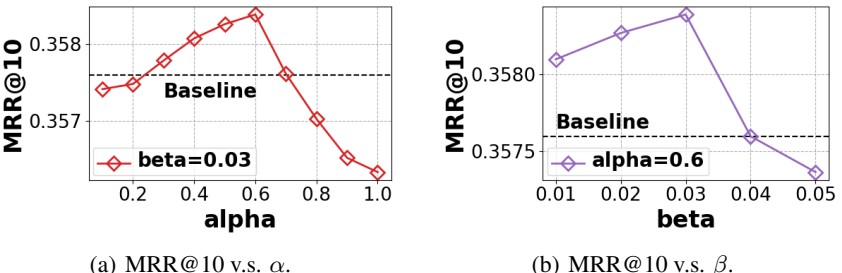

(a) MRR@10 v.s. $\alpha$.       (b) MRR@10 v.s. $\beta$.

## 5 Limitations & future works

Despite the significant contribution of MEVI, there are opportunities for further improvements to enhance the model performance. First, the twin-tower model and the sequence-to-sequence model are not jointly learned, which may not yield the optimal retrieval performance. Second, the capacity of the sequence-to-sequence model is still insufficient to cope with large-scale corpus within acceptable inference latency. In future works, these issues may be addressed by the following techniques: (a) Joint-training of RQ codebooks and two models can be explored [53, 48]. (b) Model compression and acceleration techniques can be applied, such as knowledge distillation [16], non-autoregressive generation [50].

## 6 Conclusion

In this work, we introduce a novel Model-enhanced Vector Index (MEVI), which combines the advantages of both the sequence-to-sequence autoregressive model and the twin-tower dense representation model. It can be effectively applied in real-world applications due to its ability to achieve both high recall performance and fast retrieval speed on large-scale corpus. MEVI constructs an RQ structure to hierarchically cluster large-scale documents, enabling the sequence-to-sequence model to directly generate the relevant cluster identifiers given an input query; the retrieved candidate documents in the top-k clusters are further ensembled with embedding-based retrieval results for candidates re-ranking. We empirically show that MEVI achieves better model performance than baselines on the widely used large-scale retrieval datasets MSMARCO Passage and Natural Questions.

## Acknowledgments

This work is supported by National Key R&D Program of China (2022ZD0116315) and National Natural Science Foundation of China (61832001 and U22B2037). We thank Chenyan Xiong for his participation in the early discussions of MEVI and his guidance on the use of the T5-ANCE model. We thank Hang Zhang and Yeyun Gong for their guidance on the use of the AR2 model.

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

# *Appendices for* Model-enhanced Vector Index

## A    Application in industry

We apply MEVI in a commercial advertising system. When users submit their queries, the system will retrieve the advertisements from the corpus. A successful retrieval algorithm should ensure that the retrieved advertisements satisfy two essential conditions. First, they should have a relatively high CTR (click-through rate) to maximize the number of clicks. Second, they should have strong relevance to the user's query, ensuring a good user experience that the displayed advertisements are closely related to the user intent.

**Dataset and training details.** In this scenario, a large-scale internal dataset is selected from the historical advertising logs for offline evaluation. We collect approximately 80 million query-ad pairs for training and 40 million for testing. There are approximately 20 million advertisements in the sampled dataset. To satisfy both the CTR and relevance goals, the retrieval model is followed by a relevance-targeting re-rank model to select top ads from the recall candidates. We adopt RQ codebook with 5 clustering layers and 5 bits per layer for MEVI. The model is trained on 8 V100 GPU cards for 2 weeks, with a batch size of 256. The evaluation beam size is 10. Other settings and hyperparameters keep the same as the MSMARCO Passage experiments.

**Metrics.** We adopt several metrics to evaluate recall and relevance performance. Hitrate (Hit@K) measures how many clicked ads are included in top K results given a query. MRR (MRR@K) measures the ranking performance of the ads. Average Relevance score (Rel@K) measures the relevance score from top-K results, calculated by a commercial large language model finetuned by human labels, which has been proved to be effective and correlate closely to human labels.

**Baseline.** We adopt a commercial embedding-based retrieval (EBR) methods as the baseline, which has 4 transformer layers on both query and document sides. The hidden dimension is 768 and the number of head is 12. The EBR model is trained on 8 V100 GPU cards for 6 epochs. After the training finished, we generate query and ads embeddings and adopt an efficient ANN algorithm used in production to search for relevant advertisements. We also use the same relevance model to rerank the recall candidates.

Table 1: Experiment results on commercial advertising dataset.

| Method | Hit@50 | Hit@500 | MRR@50 | MRR@500 | Rel@50 | Rel@500 |
|---|---|---|---|---|---|---|
| EBR | 61.80 | 95.23 | 73.42 | 97.07 | 39.61 | 22.92 |
| Top-10 Clusters | 69.62 | 83.40 | 75.30 | 84.35 | 26.57 | 14.98 |
| MEVI | **71.92** | **96.25** | **80.55** | **97.75** | **39.78** | **23.23** |

**Results.** The experimental results are shown in Table 1. The search results within top-10 clusters demonstrate strong recall performance for the top 50 results. However, the top 500 results show a decline in performance compared to the EBR baseline, primarily due to the beam size being limited to 10 during MEVI inference, which results in a smaller search space. Increasing the beam size could potentially improve performance but would also place a greater computation burden on the relevance model. Consequently, a beam size of 10 was ultimately chosen for production scenario.

Despite these limitations, the ensemble results reveal a substantial increase across all metrics, indicating that MEVI serves as an excellent candidate provider for recall and is complementary to embedding-based retrieval methods, which is meaningful for multiple-recall scenarios. By incorporating MEVI, the commercial advertising system can significantly enhance both recall and relevance performance.

**Online A/B test.** MEVI also demonstrates impressive results in online A/B test. It achieves a 0.71% RPM (Revenue per Mille searches) growth in the A/B test with a P-Value of 0.00021 (a P-Value less than 0.05 is considered statistically significant in production). Moreover, MEVI shows a 0.60% reduction of bad relevance cases according to the human judgement, further demonstrating that MEVI not only boost advertising revenue, but also effectively improve user experiences with better query-ad relevances.

