# OpenReview forum: "Model-enhanced Vector Index"
_NeurIPS.cc/2023/Conference — NeurIPS 2023 poster_

### Official Review · Reviewer_KfyS · 2023-06-25

**Soundness:** 3 good
**Presentation:** 3 good
**Contribution:** 3 good
**Rating:** 5
**Confidence:** 4

**Summary:**

In this study, the authors present a new approach called Model-enhanced Vector Index (MEVI) that combines ideas from both autoregressive sequence-to-sequence models for indexing and dense retrieval models (twin-tower architectures), and includes Residual Quantization. This approach offers significant advantages and proves to be highly effective in real-world applications. MEVI demonstrates nice performance gains in terms of achieving both high recall accuracy and faster retrieval speed, even when dealing with large-scale corpora.

**Strengths:**

* The method introduced combines two of the recent approaches for document retrieval - dense retrieval and the generative neural indexing. It showcases the power of these together and is extensively evaluated and experimented, both in an offline manner as usual, but also online in real-word scenarios (in commercial advertising). They also include a nice latency evaluation.
* The paper is well-written and easy to follow in general.


**Weaknesses:**

Major comments:
* MSMARCO is the only benchmark which was evaluated in this work. I’d love to see if the method generalizes well across more datasets. This is required in order to convince readers to favour this method.
* While an online A/B testing experiment appreciated, there is no guarantee about the quality of such kind of testing, while controlled crowdsourcing does provide an infrastructure for filtering out bad annotations.
* L152: I am not convinced that the tree structure is problematic because of decoding time. It can actually alleviate the decoding if using controlled decoding instead of plain beam search. Also, the hierarchical nature of the structure of embeddings proposed in this work is not that different (as stated in L167).
* The authors report results using MRR@10 Recall@50 Recall@1000 but it woul’d be interesting to see both more fine-grained retrieval results (Recall@1) following previous works, as well as a figure which states how the method results progress as k is being increased.
The training process of the twin-tower model and the sequence-to-sequence model are distinct which may be very problematic. But I do acknowledge this was stated in the limitations section.


Minor comments:
* Please elaborate more both in the contributions part in the introduction section about and in the introduction itself more about the proposed MEVI method. After reading this, I didn’t have any clue about the method which should be stated much early in the text.
* Related work section - please state how your method differs from other recent works.
* L146: insert space after T5-ANCE. Similar comment for the paragraph starting in 149 and the rest of the paper.
* L157-166: The mathematical notations lack clarity in terms of explicitly defining the dimensions of the vectors/matrices and what they represent.
* Table 4: please include detailed caption so the table can be understood standalone. The same applies for the rest of the tables.


**Questions:**

See weaknesses above.

**Limitations:**

Yes.

---

> ### Author Rebuttal · Authors · 2023-08-10
>
> Trank you for your thoughful comments and suggestions.
>
>
> We address the points you mentioned in the weaknesses part.
>
> Major comments
>
> 1. We add another dataset Natural Questions (NQ). We take AR2 (https://arxiv.org/abs/2110.03611) as the dense retriever and HNSW as the ANN algorithm. The results are listed in the table below. MEVI also achieves better performance than baseline methods on NQ dataset.
>
> | Method | R@5 | R@20 | R@100 |
> | -------- | :-----: | :-----: | :-----: |
> | BM25 | / | 59.1 | 73.7 |
> | AR2 + HNSW | 70.89 | 78.50 | 83.02 |
> ||||
> | MEVI Top-10 Clus | 59.61 | 66.45 | 71.63 |
> | MEVI Top-100 Clus | 70.33 | 77.23 | 81.77 |
> | MEVI Top-1000 Clus | 75.57 | 82.83 | 87.31 |
> ||||
> | MEVI Top-10 Clus & AR2(HNSW) | 74.10 | 82.11 | 86.43 |
> | MEVI Top-100 Clus & AR2(HNSW) | 74.43 | 82.71 | 87.51 |
> | MEVI Top-1000 Clus & AR2(HNSW) | **75.93** | **83.96** | **88.98** |
>
> 2. The online A/B test results come from our company's real online service business, which has brought higher advertising revenue with higher model quality. We present this experimental result to show that MEVI is not only applicable to academic datasets, but also effective in real industrial applications.
>
> 3. NCI's high latency does not come from the tree structure, but from its long decoding steps. We add a latency comparison between NCI and MEVI in the following table. We decouple the problem into two parts, one is the difference between MEVI and NCI/DSI, and the other is the difference between RQ and normal k-means. For the first part, each minimal cluster in MEVI contains a set of documents, while each minimal cluster in NCI and DSI contains only one document. This design makes MEVI smaller number of clustering layers, leading to smaller serving latency; also if the doc-id is too long, the model cannot learn well, so the results of MEVI is better than NCI. For the second part, we use RQ instead of normal K-Means in MEVI because RQ performs better. In RQ, each layer uses the residual vectors from the last clustering layer, and conducts k-means clustering among all the vectors. Comparing to hierarchical k-means that focuses on more fine-grained clustering within large clusters, RQ adopts residual information to address the errors of the last clustering layer, making the clustering results more robust. We conduct an experiment to compare RQ with normal hierarchical k-means. In the experiments, we test MEVI-RQ and MEVI-KMeans on MSMARCO Passage dataset, setting layer depth to 4 and the number of clusters per layer to 32. The results are listed below, showing that RQ generally achieves better recall than hierarchical k-means under the same number of layers.
>
> | Method | MRR@10 | Latency (ms) |
> | -------- | :-----: | :-----: |
> | T5-ANCE (HNSW) | 33.21 | 19.71 |
> | NCI | 26.18 | 2899.17 |
> | Top-10 & HNSW | 35.22 | 96.87 |
> | Top-100 & HNSW | 35.60 | 222.55 |
> | Top-1000 & HNSW | 35.83 | 1662.84 |
>
> | Method | MRR@10 | Recall@50 | Recall@1000 |
> | -------- | :-----: | :-----: | :-----: |
> | RQ MEVI Top-10 Clus | 32.05 | 63.25 | 66.82 |
> | RQ MEVI Top-100 Clus | 35.16 | 79.14 | 88.22 |
> | RQ MEVI Top-1000 Clus | **35.76** | **82.37** | **95.17** |
> ||||
> | K-Means MEVI Top-10 Clus | 31.62 | 65.09 | 68.25 |
> | K-Means MEVI Top-100 Clus | 34.82 | 77.30 | 87.93 |
> | K-Means MEVI Top-1000 Clus | 35.65 | 81.01 | 94.17 |
>
> 4. In the main experiments, we align with existing work (e.g. coCondenser https://arxiv.org/abs/2108.05540, RocketQA https://arxiv.org/abs/2010.08191, AR2 https://arxiv.org/abs/2110.03611) and report MRR@10, Recall@50, Recall@1000. Following your suggestion, we add Figure 2 into the attached PDF to show how the results vary with different $k$, where the ensemble method consistently outperforms T5-ANCE+HNSW. We currently separate the training of the two-tower model and the sequence-to-sequence model to eliminate the overall training cost. We plan to explore joint-training optimization in the future.
>
>
> Minor comments
>
> 1. We incorporate the information from sequence-to-sequence model into dense retrievers, achieving better model quality on large datasets with acceptable serving latency. For the generative sequence-to-sequence model, we construct an RQ codebook to cluster the documents, limiting the decoding steps as well as improving the clustering quality. For the ensembling, we take the retrieved documents from the dense retriever and the generative model, then re-rank the documents with a proper score. In experiments, we exhibits better recall performance than baseline dense retriever and generative model. We have also applied MEVI in real industrial applications, bringing higher advertising revenue.
>
> 2. Existing state-of-the-art methods can be classified into dense retrievers and generative models, where the former applies to large datasets and the latter performs better on small datasets. MEVI incorporates information from generative models into dense retrievers, achieving better model quality on large datasets.
>
> 3. Thanks for your suggestions in items 3-5. We will modify the typos and improve the captions in the next version of this paper. We have also added a table for notations, which will be placed in the appendix later.
>
>
> Notations
>
> | Notation | Explanation |
> | ----- | ----- |
> | $n$ | The number of documents |
> | $d$ | Embedding dimension |
> | $b$ | The number of bits for clustering |
> | $m$ | The number of clustering layers, i.e. the length of cluster-id |
> | $X \in \mathbb{R}^{n\times d}$ | All document embedding |
> | $B_i\in\mathbb{R}^{b\times d}, i=1,...,m$ | Centroid embeddings of clusters |
> | $B\in\mathbb{R}^{mb\times d}$ | Overall centroid embeddings |
> | $C_i\in\{0,1\}^{n\times b}, i=1,...,m$ | Code-style assignment of documents |
> | $C\in\{0,1\}^{n\times mb}$ | Overall cluster assignment |
> | $\widetilde{C}\in\{0,1,2,...,2^b-1\}^{n\times m}$ | Compact cluster assignment (typos in the original paper will be modified) |
> | $R_i\in \mathbb{R}^{n\times d},i=1,...,m-1$ | Residual embeddings after clustering |

---

> > ### Comment · Reviewer_KfyS · 2023-08-12
> >
> > I thank the authors for their responses. Although the additional works that the authors attached include more extensive evaluations, and over more than just MSMARCO and NQ, I’m willing to raise my score. In addition, I still think that A/B test doesn’t fit here, mistletoes because it is not reproducible.

---

> > > ### Author Response · Authors · 2023-08-15
> > >
> > > Thanks for your recognition of our revision. Based on your suggestions, in the next version we will move the new reproducible experiments to the main body of the paper, and the A/B test section to the appendix.

---

### Official Review · Reviewer_VZqD · 2023-07-04

**Soundness:** 3 good
**Presentation:** 3 good
**Contribution:** 3 good
**Rating:** 7
**Confidence:** 4

**Summary:**

This work introduces residual quantisation codebook into k-mean clustering to generate semantic IDs. These semantic IDs are used to provide initial cluster-level ranking and prune the corpus into document subset with thousand documents. The interpolation between the dual-encoder similarity and a derived cluster score is used as final ranking. The experiments are on msmarco passage dataset.

**Strengths:**

- An novel idea is implemented and examined to exploit the semantic ids from generative retrievers to introduce clusters into dual-encoder
- The writing is easy to follow and the method is insightful
- Promising results are reported on large in-house dataset Ms Marco and on an online system

**Weaknesses:**

- The cluster-based retrieval has been long studied and this paper should be put into context better, e.g., [1, 2] . Besides, is this the first work to introduce clusters into dual-encoder retriever?
- The effectiveness improvement in Table 1 comes from the fine-grained clustering information. Does such boost still exist when using state-of-the-art dual-encoder retrievers, like rocketqav2 and GTR, where the document similarity is with higher quality? Actually, in Table 1, it can be seen that use of HNSW could boos the recall considerably. Also, the results of the online system are also on top of weak dual-encoder models (four layers). Thus, it is beneficial to better understand when and where the clustering information could help.
- For the dynamic update of documents, is this the first work that investigates this problem? What are the existing methods tackling this problem and how they design the experiments? The results from Table 2 look promising, but I am not sure how convincing it is to support that claims of dynamic updates.
- As an ablation results, Table 3 is hard to read, and the configuration differences among rows are more than one out of three (subgroup, bits, top-k), and it is hard to answer by increasing subgroup or k it helps.
- Some formatting issues: a space is required between the text and their follow-up citation, like line 150


1. https://dl.acm.org/doi/pdf/10.1145/253495.253524
2. https://arxiv.org/abs/2008.00150

**Questions:**

- What is the configuration of \alpha and \beta and how sensitive is the performance relative to their configs?

---

> ### Author Rebuttal · Authors · 2023-08-10
>
> We appreciate the time you took to review our paper.
>
> First, we address the five points you mentioned in the weaknesses part.
>
> 1. From the taxonomy perspective, our proposed MEVI is a generation-enhanced dense retrieval method, not a traditional cluster-based retrieval method. The clustering process here forms doc-ids (or say cluster-ids) to build outputs of a generative language model. We would like to claim this is the first work to introduce information from generative model into dual-encoder retriever. We will reference the cluster-retrieval based methods and clarify their difference in the next version of our paper.
>
> 2. We have conducted experiments with another state-of-the-art dense retrieval model AR2 (https://arxiv.org/abs/2110.03611) in the table below. AR2 performs better than RocketQA-v2. The comparison of vanilla dense retriever and ensembling method in the table below shows the gain of introducing generative-based retrieval information.
>
> | Method | MRR@10 | Recall@50 | Recall@1000 |
> | -------- | :-----: | :-----: | :-----: |
> | BM25 | 18.7 | 59.2 | 85.7 |
> | T5-ANCE + HNSW | 33.21 | 77.30 | 88.61 |
> | AR2 + HNSW | 35.54 | 78.80 | 87.11 |
> | NCI | 26.18 | 74.68 | 92.44 |
> ||||
> | MEVI Top-10 Clus | 32.05 | 63.25 | 66.82 |
> | MEVI Top-100 Clus | 35.16 | 79.14 | 88.22 |
> | MEVI Top-1000 Clus | 35.76 | 82.37 | 95.17 |
> ||||
> | MEVI Top-10 Clus & T5-ANCE(HNSW) | 35.22 | 81.29 | 93.21 |
> | MEVI Top-100 Clus & T5-ANCE(HNSW) | 35.60 | 82.27 | 95.62 |
> | MEVI Top-1000 Clus & T5-ANCE(HNSW) | 35.82 | 82.77 | 97.12 |
> ||||
> | MEVI Top-10 Clus & AR2(HNSW) | 37.00 | 82.64 | 93.46 |
> | MEVI Top-100 Clus & AR2(HNSW) | 38.42 | 84.52 | 96.23 |
> | MEVI Top-1000 Clus & AR2(HNSW) | **39.16** | **86.12** | **97.65** |
>
> 3. Dynamic update is not a problem in dense retrieval, since the usage of embedding already enables unseen documents if the new-coming ones do not change the overall distribution drastically. However, in existing generation-based methods which convert documents to fixed doc-ids, re-training is required if new documents come in. Therefore, existing dense retrieval methods have better ability to handle new documents, while existing generation-based methods cannot support dynamic update. Since we aim to incorporate the information from generative models and take the advantage of dense retrieval methods simultaneously, we train the model with only 90% documents and regard the rest as new documents to validate the capability of MEVI to handle new documents. In experiments, MEVI has comparable results to dense retrieval methods, exhibiting similar ability to support dynamic updates.
>
> 4. In Table 3 of the original paper, our aim is to examine the recall performance of different configurations with almost the same number of candidate documents per query. To align the magnitude of the number of documents per query, we have to adjust the number of clusters, which is calculated as (2 ^ #subgroup) ^ #bits. The numbers of subgroup and bits both determine the size of cluster-id space, while the k determines the number of documents per query. From RQ(3,4) to RQ(5,5), the number of clusters increases, and the number of documents in each cluster decreases; so we enlarge $k$ to align the magnitude of the number of documents per query. In general, the larger number of candidate documents, the more likely the model is to recall the correct document. Comparing RQ(3,4), RQ(4,4), and RQ(4,5), even the number of documents per query is reduced, the recall performance increases due to finer-grained learning to generate cluster-ids. In RQ(5,5) setting, the cost of generation becomes unacceptable as the number of clusters increases. In summary, RQ(4,5) achieves the best trade-off between model performance and cluster-id space size.
>
> 5. Thanks for your valuable suggestions. We will modify the typos in the next version.
>
> Then we answer the questions below.
>
> 1. What is the configuration of $\alpha$ and $\beta$ and how sensitive is the performance relative to their configs?
> - $\alpha$ determines the overall contribution ratio of clustering information, while $\beta$ determines how large the clustering score is with respect to the cluster rank. In practice, we directly search and choose the best configurations since the ensembling does not incur additional training or inference, and the search overhead can be ignored. Figure 1 in the attached PDF shows the MRR@10 for different values of $\alpha$ and $\beta$. With the right configuration, the ensemble results achieves optimal performance as it fully leverages the two components. The best hyperparameters searched on the dev set are also optimal on the test set.

---

> > ### Comment · Reviewer_VZqD · 2023-08-17
> >
> > I thank the authors for the detailed response and the additional results. I am willing to increase my rating.
> >
> > For 1 and 2, that makes sense to me. It seems with AR2 the performance gets even better (more boost?).
> >
> > For 3 and 4, thanks for the explanation and that makes sense. For 3, the update is indeed an issue since it is hard for the gen model to provide cluster id to the unseen doc, but I think it is not of concern for me and the 90% experiment looks good.
> >
> > For the attache pdf, can you claim the robustness for different choices of hyper-parameters from the results (the range of y-axis is relatively small)?

---

> > > ### Author Response · Authors · 2023-08-17
> > >
> > > Thanks for your recognition of our revision.
> > >
> > > MEVI achieves significant gains on both T5-ANCE and AR2 retrievers. The performance gain of MEVI+AR2 is larger, probably because AR2 is a more effective retriever with a higher upper bound, and the information it contained is fully utilized in the ensemble process.
> > >
> > > Our ensemble method is robust to hyperparameters. As shown in Figure 1, the ensemble result outperforms the baseline (35.76) when $0.3\le\alpha\le0.7$ and $0.01\le\beta\le0.03$. This finding will be added into the next version of our paper.

---

### Official Review · Reviewer_aDGQ · 2023-07-06

**Soundness:** 3 good
**Presentation:** 3 good
**Contribution:** 2 fair
**Rating:** 6
**Confidence:** 5

**Summary:**

This paper proposes to improve dense neural IR models by adding a RQ structure (hierarchical clustering based on residuals at each step) before the ANN search. The structure is used to construct a semantic identifier string for each document. The authors thus use a generative approach to retrieval combined with a dense model.

This approach has two advantages, firstly to have a faster retrieval (since the ANN search is restricted to a subset of the document), secondly to have a better retrieval (by exploiting a cluster-based score). Experiments conducted on MS Marco (passages) and a proprietary advertisement dataset show improvements over T5-ANCE.

**Strengths:**

The paper is well written and technical details ensuring reproduction are given.

The approach is promising as it both allows for more effective and efficient retrieval - which is always a challenge in IR. The proposed approach, based on RQ clusters, is original and combines the strength of generative and dense models in IR.

It also shows that adding documents is not as problematic as for other generative approaches (but however the performance is worse than T5-ANCE in that case, for a higher latency).

**Weaknesses:**

- Comparison with state of the art models (A2R/Adversarial Retriever-Ranker for dense models, SPLADE for sparse ones) is missing - it would have been good to see how with better dense models the proposed approach evolves.

- With respect to generative models, several approaches have better results than NCI nowadays (e.g. Ultron) or the more recent “Learning to tokenise” (SIGIR’23). At least Ultron should be reported (ArXiv 2022) for comparison purposes.

- The authors have chosen to use a proprietary dataset (section 4.7) for the second evaluation. It would have been much wiser to use publicly available ones as nothing can be checked on this one (e.g. BEIR) - reporting proprietary results in the appendix would have been appropriate.

**Questions:**

- what is the performance of ANCE without HNSW? This would have been a proper baseline.

- in section 3.2, how are $\alpha$ and $\beta$ set?

- What it the advantage of using RQ over hierarchical clustering?

**Limitations:**

The limitation section contradicts a bit the initial claims (“… the capacity of the sequence-to-sequence model is still insufficient to cope with large-scale corpus within acceptable inference latency …”), but correctly lists the different limitations of the paper.

---

> ### Author Rebuttal · Authors · 2023-08-10
>
> Thank you for your detailed review.
>
> First, we address the three points you mentioned in the weaknesses part.
>
> 1. On the MSMARCO Passage dataset, we add the experiment results with another state-of-the-art dense retrieval model AR2 (https://arxiv.org/abs/2110.03611) in the following table. We also add SPLADE results for comparison. MEVI achieves better performance than AR2 (with HNSW as the ANN algorithm). Note that AR2 without HNSW performs brute force to rank all candidates, which gets better results but is not applicable in online applications.
>
> | Method | MRR@10 | Recall@50 | Recall@1000 |
> | -------- | :-----: | :-----: | :-----: |
> | BM25 | 18.7 | 59.2 | 85.7 |
> | SPLADE | 32.2 | / | 95.5 |
> | T5-ANCE | 35.73 | 82.96 | 97.21 |
> | T5-ANCE + HNSW | 33.21 | 77.30 | 88.61 |
> | AR2 | 39.50 | 86.98 | 98.44 |
> | AR2 + HNSW | 35.54 | 78.80 | 87.11 |
> | NCI | 26.18 | 74.68 | 92.44 |
> ||||
> | MEVI Top-10 Clus | 32.05 | 63.25 | 66.82 |
> | MEVI Top-100 Clus | 35.16 | 79.14 | 88.22 |
> | MEVI Top-1000 Clus | 35.76 | 82.37 | 95.17 |
> ||||
> | MEVI Top-10 Clus & T5-ANCE(HNSW) | 35.22 | 81.29 | 93.21 |
> | MEVI Top-100 Clus & T5-ANCE(HNSW) | 35.60 | 82.27 | 95.62 |
> | MEVI Top-1000 Clus & T5-ANCE(HNSW) | 35.82 | 82.77 | 97.12 |
> ||||
> | MEVI Top-10 Clus & AR2(HNSW) | 37.00 | 82.64 | 93.46 |
> | MEVI Top-100 Clus & AR2(HNSW) | 38.42 | 84.52 | 96.23 |
> | MEVI Top-1000 Clus & AR2(HNSW) | 39.16 | 86.12 | 97.65 |
>
> 2. Ultron conducts experiments on MSMARCO DOCUMENT and NQ-320K datasets, which has much fewer documents than MSMARCO PASSAGE and NQ datasets (we add experiments of this dataset in the second table) in our experiments. Due to the high decoding cost of generative models, Ultron is not friendly to larger datasets. Therefore, we do not compare to Ultron in our experiments. Moreover, the results presented in the Ultron paper are no better than NCI. For another paper you mentioned as "learning to generate", we can not find this paper in both Arxiv and SIGIR'23 venue. Could you provide a link to this paper for our reference?
>
> 3. We conduct experiments on another public dataset Natural Questions (NQ) and the results are listed in the table below. MEVI still achieves better performance than baseline methods.
>
> | Method | R@5 | R@20 | R@100 |
> | -------- | :-----: | :-----: | :-----: |
> | BM25 | / | 59.1 | 73.7 |
> | AR2 | 77.78 | 85.98 | 90.03 |
> | AR2 + HNSW | 70.89 | 78.50 | 83.02 |
> ||||
> | MEVI Top-10 Clus | 59.61 | 66.45 | 71.63 |
> | MEVI Top-100 Clus | 70.33 | 77.23 | 81.77 |
> | MEVI Top-1000 Clus | 75.57 | 82.83 | 87.31 |
> ||||
> | MEVI Top-10 Clus & AR2(HNSW) | 74.10 | 82.11 | 86.43 |
> | MEVI Top-100 Clus & AR2(HNSW) | 74.43 | 82.71 | 87.51 |
> | MEVI Top-1000 Clus & AR2(HNSW) | 75.93 | 83.96 | 88.98 |
>
> Then we answer the questions below.
>
> 1. What is the performance of ANCE without HNSW? This would have been a proper baseline.
> - We add the performance of ANCE without HNSW in the tables above. Though the ideal result of ANCE achieves good recall performance especially when k is large, it is not applicable in real industrial scenarios due to large serving latency (> 3500 ms), so we do not explicitly include it in our paper. MEVI can reach better or nearly the same model quality with less serving latency.
>
>
>
> 2. In section 3.2, how are $\alpha$ and $\beta$ set?
> - We search $\alpha$ and $\beta$ within a proper range and set them according to the model metric. Since the ensemble process does not incur additional training and inference, the time for search can be ignored. Figure 1 in the attached PDF shows the MRR@10 for different values of $\alpha$ and $\beta$. With the right configuration, the ensemble results achieves optimal performance as it fully leverages both components. In general, for the same dataset, the best hyperparameters searched on the dev set are also optimal on the test set.
>
> 3. What it the advantage of using RQ over hierarchical clustering?
> - In RQ, each layer uses the residual vectors from the last clustering layer, and conducts k-means clustering among all the vectors. Comparing to hierarchical k-means that focuses on more fine-grained clustering within large clusters, RQ adopts residual information to address the errors of the last clustering layer, making the clustering results more robust. We conduct an experiment to compare RQ with normal hierarchical k-means. In the experiments, we test MEVI-RQ and MEVI-KMeans on MSMARCO Passage dataset, setting layer depth to 4 and the number of clusters per layer to 32. The results are listed in the table below, showing that RQ generally achieves better recall than hierarchical k-means under the same number of layers.
>
> | Method | MRR@10 | Recall@50 | Recall@1000 |
> | -------- | :-----: | :-----: | :-----: |
> | RQ MEVI Top-10 Clus | 32.05 | 63.25 | 66.82 |
> | RQ MEVI Top-100 Clus | 35.16 | 79.14 | 88.22 |
> | RQ MEVI Top-1000 Clus | **35.76** | **82.37** | **95.17** |
> ||||
> | K-Means MEVI Top-10 Clus | 31.62 | 65.09 | 68.25 |
> | K-Means MEVI Top-100 Clus | 34.82 | 77.30 | 87.93 |
> | K-Means MEVI Top-1000 Clus | 35.65 | 81.01 | 94.17 |

---

> > ### Comment · Reviewer_aDGQ · 2023-08-14
> >
> > Many thanks for the answers – I think the complementary results are nice (there was no answer for the use of a proprietary dataset)./
> >
> > Few other notes:
> >
> > - SPLADE has different versions – the one you report is v1 and is far from optimal
> > - efficiency results should be reported along with effectiveness ones (when applicable)
> > - the "learning to tokenize" (sorry, not "to generate") paper: http://arxiv.org/abs/2304.04171

---

> > > ### Author Response · Authors · 2023-08-15
> > >
> > > Thanks for your recognition of our revision. We make some supplementary explanations on the above questions.
> > >
> > > 1. Currently we have used two large-scale public datasets, MSMARCO Passage and Natural Questions, which are commonly used in related works such as coCondenser, RocketQA, and AR2. They are also adopted in BEIR benckmark (NQ in BEIR is smaller; here we use the version with large-scale documents as in previous works). In the next version, we will place the experiments on public datasets in the paper, and move the A/B test section to appendix.
> > >
> > > 2. We add the results of SPLADE-v2 to the table of MSMARCO Passage below.
> > >
> > > | Method | MRR@10 | Recall@50 | Recall@1000 |
> > > | -------- | :-----: | :-----: | :-----: |
> > > | BM25 | 18.7 | 59.2 | 85.7 |
> > > | SPLADE | 32.2 | / | 95.5 |
> > > | SPLADE-v2 | 36.8 | / | 97.9 |
> > > | T5-ANCE | 35.73 | 82.96 | 97.21 |
> > > | T5-ANCE + HNSW | 33.21 | 77.30 | 88.61 |
> > > | AR2 | 39.50 | 86.98 | 98.44 |
> > > | AR2 + HNSW | 35.54 | 78.80 | 87.11 |
> > > | NCI | 26.18 | 74.68 | 92.44 |
> > > ||||
> > > | MEVI Top-10 Clus | 32.05 | 63.25 | 66.82 |
> > > | MEVI Top-100 Clus | 35.16 | 79.14 | 88.22 |
> > > | MEVI Top-1000 Clus | 35.76 | 82.37 | 95.17 |
> > > ||||
> > > | MEVI Top-10 Clus & T5-ANCE(HNSW) | 35.22 | 81.29 | 93.21 |
> > > | MEVI Top-100 Clus & T5-ANCE(HNSW) | 35.60 | 82.27 | 95.62 |
> > > | MEVI Top-1000 Clus & T5-ANCE(HNSW) | 35.82 | 82.77 | 97.12 |
> > > ||||
> > > | MEVI Top-10 Clus & AR2(HNSW) | 37.00 | 82.64 | 93.46 |
> > > | MEVI Top-100 Clus & AR2(HNSW) | 38.42 | 84.52 | 96.23 |
> > > | MEVI Top-1000 Clus & AR2(HNSW) | 39.16 | 86.12 | 97.65 |
> > >
> > >
> > > 3. We conduct experiments of efficiency in another experiments, and the results are shown below.
> > >
> > > | Method | MRR@10 | Latency (ms) |
> > > | -------- | :-----: | :-----: |
> > > | T5-ANCE (HNSW) | 33.21 | 19.71 |
> > > | NCI | 26.18 | 2899.17 |
> > > | MEVI Top-10 & HNSW | 35.22 | 96.87 |
> > > | MEVI Top-100 & HNSW | 35.60 | 222.55 |
> > > | MEVI Top-1000 & HNSW | 35.83 | 1662.84 |
> > >
> > >
> > > 4. We notice that "learning to tokenize" is a preprint paper which still works in progress, and their codes are not open-sourced. Thus we are not able to conduct experiments based on it. It uses NQ-320K and sampled MSMARCO Passage datasets, which are much smaller than NQ and full MSMARCO Passage datasets in our experiments, so the results are not comparable. Since the components in MEVI are highly decoupled, we believe that MEVI is also applicable to generative models other than NCI, which we consider as future work.

---

> > > > ### Comment · Reviewer_aDGQ · 2023-08-22
> > > >
> > > > Thanks for these further details and comments; I updated my review to take into account the different additional results provided by the authors.

---

### Official Review · Reviewer_c7Dg · 2023-07-09

**Soundness:** 3 good
**Presentation:** 3 good
**Contribution:** 3 good
**Rating:** 6
**Confidence:** 5

**Summary:**

This paper present a new method for ensembling generative retrieval and embedding-based dense retrieval. Prior generative retrieval work that solely relies on a generation model to generate the retrieval document, but it is difficult to scale to large corpus. In contrast, this paper uses the generation model to predict candidate document clusters, but still adopt a dense retriever to produce the final document rankings, where the document from different clusters are weighted according the generation model's output.

The authors first evaluated this method on MS MARCO. The proposed method show improvements over a dense retriever T5-ANCE, and also good support to corpus update. The authors also tested this method in a production ad system where improvements were also observed over their production dense retrieval model (a 4-layer transformer dual-encoder).

**Strengths:**

- This paper proposes a simple idea to ensemble generative retrieval model and embedding-based dense retrieval.  In the proposed method, the generative retrieval model is essentially predicting document clusters from a query which are then used for reweighting or pruning the dense retrieval results. This aproach is a lot more practical and scalable compared to existing generative retrieval approaches that need to generate the exact document identifier.
- Experiments show promising results.
- Paper is well written.

**Weaknesses:**

- The authors did not compare to any published neural retrieval baselines on MS MARCO, as T5-ANCE and NCI are both their own implementations. I'm wondering how does the method work with SOTA dense retrievers such as RocketQA v2 or coCondensor, whose results seem stronger than T5-ANCE on MS MARCO.
- It would be nice to add another dataset, such as Natural Questions.
- It is unclear to me what is the advantage of RQ compared to other hierarchical clustering approaches. It would be nice to add some discussion and ablation studies.
- There are some confusions around T5-ANCE. First of all, T5-ANCE does not seem to exactly follow the ANCE recipe which uses dynamic indexing for hard negative mining. Second, it is unclear the model size and hyperparameters for this model.
- I think is is an over-claim to say "For the first time, we demonstrate that a novel-designed generation-based model is able to handle
 a large corpus with millions of documents, reaching high recall performance and low serving latency at the same time." From my understanding, this method mostly rely on dense retrieval and generation-based model is only providing cluster-level weights. To justify this claim, the author should report the performance without ensembling with dense retrieval.

**Questions:**

- How does the model compares or works with SOTA dense retrievers on MS MARCO?
- What is model size of the generation model? How does it affect serving latency?
- How's the performance with just the generation model?

---

> ### Author Rebuttal · Authors · 2023-08-10
>
> Thank you for your valuable feedback.
>
> First, we address the five points you mentioned in the weaknesses part.
>
> 1. On the MSMARCO Passage dataset, we add another state-of-the-art dense retrieval model AR2 (https://arxiv.org/abs/2110.03611) in addition to T5-ANCE. AR2 performs better than RocketQA-v2 and coCondenser. From the experiment results in the table below, we can see that MEVI also achieves a better performance than AR2+HNSW.
>
> | Method | MRR@10 | Recall@50 | Recall@1000 |
> | -------- | :-----: | :-----: | :-----: |
> | BM25 | 18.7 | 59.2 | 85.7 |
> | T5-ANCE + HNSW | 33.21 | 77.30 | 88.61 |
> | AR2 + HNSW | 35.54 | 78.80 | 87.11 |
> | NCI | 26.18 | 74.68 | 92.44 |
> ||||
> | MEVI Top-10 Clus | 32.05 | 63.25 | 66.82 |
> | MEVI Top-100 Clus | 35.16 | 79.14 | 88.22 |
> | MEVI Top-1000 Clus | 35.76 | 82.37 | 95.17 |
> ||||
> | MEVI Top-10 Clus & T5-ANCE(HNSW) | 35.22 | 81.29 | 93.21 |
> | MEVI Top-100 Clus & T5-ANCE(HNSW) | 35.60 | 82.27 | 95.62 |
> | MEVI Top-1000 Clus & T5-ANCE(HNSW) | 35.82 | 82.77 | 97.12 |
> ||||
> | MEVI Top-10 Clus & AR2(HNSW) | 37.00 | 82.64 | 93.46 |
> | MEVI Top-100 Clus & AR2(HNSW) | 38.42 | 84.52 | 96.23 |
> | MEVI Top-1000 Clus & AR2(HNSW) | **39.16** | **86.12** | **97.65** |
>
> 2. We conduct experiments on Natural Questions (NQ) and the results are listed here. We take AR2 as the dense retriever and HNSW as the ANN algorithm. MEVI achieves better performance than baseline methods on NQ.
>
> | Method | R@5 | R@20 | R@100 |
> | -------- | :-----: | :-----: | :-----: |
> | BM25 | / | 59.1 | 73.7 |
> | AR2 + HNSW | 70.89 | 78.50 | 83.02 |
> ||||
> | MEVI Top-10 Clus | 59.61 | 66.45 | 71.63 |
> | MEVI Top-100 Clus | 70.33 | 77.23 | 81.77 |
> | MEVI Top-1000 Clus | 75.57 | 82.83 | 87.31 |
> ||||
> | MEVI Top-10 Clus & AR2(HNSW) | 74.10 | 82.11 | 86.43 |
> | MEVI Top-100 Clus & AR2(HNSW) | 74.43 | 82.71 | 87.51 |
> | MEVI Top-1000 Clus & AR2(HNSW) | **75.93** | **83.96** | **88.98** |
>
> 3. We find RQ performs better than normal hierarchical K-Means. In RQ, each layer uses the residual vectors from the last clustering layer, and conducts k-means clustering among all the vectors. Comparing to hierarchical k-means that focuses on more fine-grained clustering within large clusters, RQ adopts residual information to address the errors of the last clustering layer, making the clustering results more robust. We conduct an experiment to compare RQ with normal hierarchical k-means. In the experiments, we test MEVI-RQ and MEVI-KMeans on MSMARCO Passage dataset, setting cluster depth to 4 and the number of clusters per layer to 32. The results are listed in the table below, showing that RQ generally achieves better recall than hierarchical k-means under the same number of layers.
>
> | Method | MRR@10 | Recall@50 | Recall@1000 |
> | -------- | :-----: | :-----: | :-----: |
> | MEVI-RQ Top-10 Clus | 32.05 | 63.25 | 66.82 |
> | MEVI-RQ Top-100 Clus | 35.16 | 79.14 | 88.22 |
> | MEVI-RQ Top-1000 Clus | **35.76** | **82.37** | **95.17** |
> ||||
> | MEVI-KMeans Top-10 Clus | 31.62 | 65.09 | 68.25 |
> | MEVI-KMeans Top-100 Clus | 34.82 | 77.30 | 87.93 |
> | MEVI-KMeans Top-1000 Clus | 35.65 | 81.01 | 94.17 |
>
> 4. T5-ANCE is another model released by the authors of ANCE. The hyperparameters and the whole training process is shown in this website https://openmatch.readthedocs.io/en/latest/models/t5-ance.html. T5-ANCE follows the two-round training, where the first round uses BM25 negative samples and the second round uses hard negative samples. It uses T5-base as the backbone model. More detailed hyperparameters can be found in the website above.
>
> 5. Our expression here is a little bit confusing. We will revise it in the next version. We would like to demonstrate that MEVI is the first generation-ENHANCED model to handle a large corpus with millions of documents, with high recall performance and acceptable serving latency. The performance without ensembling has already been reported in Table 1 from the original version, named "MEVI Top-K Clus". The recall and MRR metrics are generally better than T5-ANCE+HNSW when K is 100 or 1000.
>
>
> Then here are the answers to your questions.
>
> 1. How does the model compares or works with SOTA dense retrievers on MS MARCO?
> - Already presented in weaknesses part. We add an experiment with sota dense retriever AR2 in the first table.
>
> 2. What is model size of the generation model? How does it affect serving latency?
> - Currently we use T5-base as the backbone generation model. We add an experiment on the serving latency of different model sizes in the following table. When the model size becomes larger, the latency becomes unacceptable for online serving.
>
> | Method | Latency (ms) |
> | -------- | :-----: |
> | T5-base Top-10 | 96.87 |
> | T5-base Top-100 | 222.55 |
> | T5-base Top-1000 | 1662.84 |
> | T5-large Top-10 | 124.56 |
> | T5-large Top-100 | 253.23 |
> | T5-large Top-1000 | 1665.47 |
>
> 3. How's the performance with just the generation model?
> - The performance is presented in the first table above, named "MEVI Top-k Clus". The generation model already achieves good recall performance without ensembling when K is 1000.

---

> > ### Comment · Reviewer_c7Dg · 2023-08-20
> >
> > Thanks for the detailed response and the additional result! These are good results and should be put into the paper. I'm willing to raise my rating to 6.

---

### Official Review · Reviewer_v1wc · 2023-07-12

**Soundness:** 2 fair
**Presentation:** 3 good
**Contribution:** 3 good
**Rating:** 4
**Confidence:** 3

**Summary:**

This research paper introduces a deep text retrieval model that possesses the capability to effectively manage a large corpus comprising millions of documents. The model achieves remarkable recall performance while maintaining relatively low latency. Moreover, in addition to surpassing the performance of existing methods in a document retrieval benchmark, the paper also showcases the successful integration of deep text retrieval models into a commercial advertising system, demonstrating their practical value in an industrial setting.

**Strengths:**

1.	S1: They are supposed to be the first to demonstrate the successful implementation of deep text retrieval models in the practical application.
2.	S2: They investigate and test the effectiveness of Residual Quantization within the deep text retrieval model.
3.	S3: The experiment results on MSMARCO dataset display the effectiveness of the proposed method.


**Weaknesses:**

W1: They conduct experiments on a single dataset for their study. However, it is recommended that their model undergo experiments on commonly used datasets such as Natural Question, which is utilized in both DSI and NCI research. In addition, they point out that the existing methods are limited by unacceptable serving latency but they do not conduct such experiments to prove this. Since their method use Residual
Quantization (RQ) as their clustering method, this method may be more time-consuming than other clustering methods during training.


**Questions:**

1.	In dynamic update experiments, if more documents come in, their codebook may not be "optimal", so how can the codebook be adjusted to preserve similar effectiveness and robustness? Otherwise, the resulting method may not be good. In addition, their Top-10 results are worse than T5-ANCE.
2.	More experiments are needed. They should also compare their method with one another datasets used in DSI or NCI. And the latency experiment should include other baseline models.
3.	If RQ is applied to DSI and NSI, does their method achieve better performance or does the utilization of normal k-means in the proposed method potentially diminish its effectiveness to a significant extent?


**Limitations:**

The authors pointed out the limitations of their model including not jointly learning between twin-tower model and seq2seq model, and their method is still unacceptable inference latency in the large-scale corpus. They had provided potential solutions to address the above issues.

---

> ### Author Rebuttal · Authors · 2023-08-10
>
> Thank you for taking the time to review our paper.
>
> First, we address the points you mentioned in the weaknesses part.
>
> 1. We conduct experiments on another popular dataset, Natural Questions (NQ). We take AR2 (https://arxiv.org/abs/2110.03611) as the dense retriver and HNSW as the ANN algoritm. As shown in the table below, MEVI method still achieves better performance than baseline methods.
>
> | Method | R@5 | R@20 | R@100 |
> | -------- | :-----: | :-----: | :-----: |
> | BM25 | / | 59.1 | 73.7 |
> | AR2 + HNSW | 70.89 | 78.50 | 83.02 |
> ||||
> | MEVI Top-10 Clus | 59.61 | 66.45 | 71.63 |
> | MEVI Top-100 Clus | 70.33 | 77.23 | 81.77 |
> | MEVI Top-1000 Clus | 75.57 | 82.83 | 87.31 |
> ||||
> | MEVI Top-10 Clus & AR2(HNSW) | 74.10 | 82.11 | 86.43 |
> | MEVI Top-100 Clus & AR2(HNSW) | 74.43 | 82.71 | 87.51 |
> | MEVI Top-1000 Clus & AR2(HNSW) | **75.93** | **83.96** | **88.98** |
>
> 2. We add the latency of NCI here for comparison. It is worth noting that the clustering process is performed in advance, while the training process and the serving process are performed on the constant RQ structure. The service latency here is mainly determined by the number of decoding steps in the autoregressive generation. Since MEVI allows each doc-id to represent a set of documents, while NCI only allows one doc-id to represent one document, MEVI has much fewer decoding steps than NCI's 10 decoding steps, resulting in a smaller latency.
>
> | Method | MRR@10 | Latency (ms) |
> | -------- | :-----: | :-----: |
> | T5-ANCE (HNSW) | 33.21 | 19.71 |
> | NCI | 26.18 | 2899.17 |
> | MEVI Top-10 & HNSW | 35.22 | 96.87 |
> | MEVI Top-100 & HNSW | 35.60 | 222.55 |
> | MEVI Top-1000 & HNSW | 35.83 | 1662.84 |
>
> Then we answer the questions below.
>
> 1. In dynamic update experiments, if more documents come in, their codebook may not be "optimal", so how can the codebook be adjusted to preserve similar effectiveness and robustness? Otherwise, the resulting method may not be good. In addition, their Top-10 results are worse than T5-ANCE.
> - If more documents come in and the distribution of documents has changed drastically, all existing dense retrieval methods, as well as MEVI, have to re-train the model and re-construct the document index. In other words, dynamic updates only support a small number of new documents that do not change the distribution much. In future work we may extend RQ to an adaptive structure that captures the distribution of new documents during training. When MEVI only searches documents within top-10 clusters, the number of documents to be ranked is very small, leading to bad recall performance. After enlarged to top-100 or top-1000 clusters, MEVI performs better than the baseline in both basic experiment and dynamic update experiment.
>
> 2. More experiments are needed. They should also compare their method with one another datasets used in DSI or NCI. And the latency experiment should include other baseline models.
> - This question has been addressed in the weakness part. The experiment results on NQ dataset (full document set version from DPR) are listed in the first table. MEVI performs better than baseline models. For latency comparison, the results are listed in the second table. Dense retrieval based on bi-encoders has the smallest latency. MEVI outperforms significantly better than NCI while holding affordable latency.
>
>
> 3. If RQ is applied to DSI and NSI, does their method achieve better performance or does the utilization of normal k-means in the proposed method potentially diminish its effectiveness to a significant extent?
> - We would like to break this problem down into two parts, one is the difference between MEVI and NCI/DSI, and the other is the difference between RQ and normal k-means. For the first difference, each minimal cluster in MEVI contains a set of documents, while each minimal cluster in NCI and DSI contains only one document. This design makes MEVI smaller number of clustering layers, leading to lower serving latency. Also, if the doc-id is too long, the model cannot learn effectively in a certain time budget, so the results of MEVI on MSMARCO Passage is much better than NCI. For the second difference, we use RQ instead of normal K-Means in MEVI because RQ performs better. In RQ, each layer uses the residual vectors from the last clustering layer, and conducts k-means clustering among all the vectors. Comparing to hierarchical k-means that focuses on more fine-grained clustering within large clusters, RQ adopts residual information to address the errors of the last clustering layer, making the clustering results more robust. We conduct an experiment to compare RQ with normal hierarchical k-means. In the experiments, we test MEVI-RQ and MEVI-KMeans on MSMARCO Passage dataset, setting layer depth to 4 and the number of clusters per layer to 32. The results are listed in the following table, showing that RQ generally achieves better recall than hierarchical k-means under the same number of layers (corresponding to comparable latency).
>
>
> | Method | MRR@10 | Recall@50 | Recall@1000 |
> | -------- | :-----: | :-----: | :-----: |
> | MEVI-RQ Top-10 Clus | 32.05 | 63.25 | 66.82 |
> | MEVI-RQ Top-100 Clus | 35.16 | 79.14 | 88.22 |
> | MEVI-RQ Top-1000 Clus | **35.76** | **82.37** | **95.17** |
> ||||
> | MEVI-KMeans Top-10 Clus | 31.62 | 65.09 | 68.25 |
> | MEVI-KMeans Top-100 Clus | 34.82 | 77.30 | 87.93 |
> | MEVI-KMeans Top-1000 Clus | 35.65 | 81.01 | 94.17 |

---

### Author Rebuttal · Authors · 2023-08-10

Dear reviewers,

Thank you for taking time in reading our paper and providing valuable comments. We briefly address common questions here.

1. We add another dataset, Natural Questions (full document set version from DPR https://arxiv.org/abs/2004.04906), for comparison. We take AR2 (https://arxiv.org/abs/2110.03611) as the dense retriver and HNSW as the ANN algoritm. As shown in the table below, MEVI performs better than  baseline methods.

| Method | R@5 | R@20 | R@100 |
| -------- | :-----: | :-----: | :-----: |
| BM25 | / | 59.1 | 73.7 |
| AR2 + HNSW | 70.89 | 78.50 | 83.02 |
||||
| MEVI Top-10 Clus | 59.61 | 66.45 | 71.63 |
| MEVI Top-100 Clus | 70.33 | 77.23 | 81.77 |
| MEVI Top-1000 Clus | 75.57 | 82.83 | 87.31 |
||||
| MEVI Top-10 Clus & AR2(HNSW) | 74.10 | 82.11 | 86.43 |
| MEVI Top-100 Clus & AR2(HNSW) | 74.43 | 82.71 | 87.51 |
| MEVI Top-1000 Clus & AR2(HNSW) | **75.93** | **83.96** | **88.98** |


2. We would like to clarify the difference between MEVI and NCI/DSI, and the difference between RQ and k-means for identifier generation. For the first difference, each minimal cluster in MEVI contains a set of documents, while each minimal cluster in NCI and DSI contains only one document. This design makes MEVI smaller number of decoding steps, leading to smaller serving latency. Also, if the doc-id is too long, the model cannot be trained effectively, so the results of MEVI on MSMARCO Passage dataset is better than NCI and has lower serving latency. For the second difference, we choose RQ instead of K-Means in MEVI for its better performance. In RQ, each layer uses the residual vectors from the last clustering layer and conducts k-means clustering among all the vectors. Comparing to hierarchical k-means that focuses on more fine-grained clustering within large clusters in each layer, RQ adopts residual information to address the errors of the previous layer, making the clustering results more precise and robust. We conduct an experiment to compare RQ with normal hierarchical k-means. In the experiments, we test MEVI-RQ and MEVI-KMeans on MSMARCO Passage dataset, setting layer depth to 4 and the number of clusters per layer to 32. The results are listed in the following table, showing that RQ generally achieves better recall than hierarchical k-means under the same number of layers.

| Method | MRR@10 | Recall@50 | Recall@1000 |
| -------- | :-----: | :-----: | :-----: |
| MEVI-RQ Top-10 Clus | 32.05 | 63.25 | 66.82 |
| MEVI-RQ Top-100 Clus | 35.16 | 79.14 | 88.22 |
| MEVI-RQ Top-1000 Clus | **35.76** | **82.37** | **95.17** |
||||
| MEVI-KMeans Top-10 Clus | 31.62 | 65.09 | 68.25 |
| MEVI-KMeans Top-100 Clus | 34.82 | 77.30 | 87.93 |
| MEVI-KMeans Top-1000 Clus | 35.65 | 81.01 | 94.17 |

3. On the MSMARCO Passage dataset, we add an experiment on the state-of-the-art dense retriever AR2 (https://arxiv.org/abs/2110.03611) in addition to T5-ANCE. From the result table below, we can see that MEVI+AR2 achieves a better performance than the corresponding baselines.

| Method | MRR@10 | Recall@50 | Recall@1000 |
| -------- | :-----: | :-----: | :-----: |
| BM25 | 18.7 | 59.2 | 85.7 |
| T5-ANCE + HNSW | 33.21 | 77.30 | 88.61 |
| AR2 + HNSW | 35.54 | 78.80 | 87.11 |
| NCI | 26.18 | 74.68 | 92.44 |
||||
| MEVI Top-10 Clus | 32.05 | 63.25 | 66.82 |
| MEVI Top-100 Clus | 35.16 | 79.14 | 88.22 |
| MEVI Top-1000 Clus | 35.76 | 82.37 | 95.17 |
||||
| MEVI Top-10 Clus & T5-ANCE(HNSW) | 35.22 | 81.29 | 93.21 |
| MEVI Top-100 Clus & T5-ANCE(HNSW) | 35.60 | 82.27 | 95.62 |
| MEVI Top-1000 Clus & T5-ANCE(HNSW) | 35.82 | 82.77 | 97.12 |
||||
| MEVI Top-10 Clus & AR2(HNSW) | 37.00 | 82.64 | 93.46 |
| MEVI Top-100 Clus & AR2(HNSW) | 38.42 | 84.52 | 96.23 |
| MEVI Top-1000 Clus & AR2(HNSW) | **39.16** | **86.12** | **97.65** |

4. To determine the values of $\alpha$ and $\beta$, we search within a proper range and set them according to the model metrics. Since the ensemble process does not incur additional training and inference cost, the time for grid search can be ignored. We show the MRR@10 for different values of $\alpha$ and $\beta$ in the attached PDF. With the best configuration, the ensemble results achieves optimal performance as it fully leverages both components. On MSMARCO dataset, the best hyperparameters searched on the dev set are also optimal on the test set.

---

### Decision · Program_Chairs · 2023-09-21

**Decision:**

Accept (poster)

**Comment:**

This is a well-written paper that proposes a highly novel way to combine generative retrieval and dense retrieval. The experimental results are strong and convincing.

The reviewers did raise a few concerns about this paper, but the most critical aspects of those were adequately addressed during the rebuttal.

Given that the strengths of this paper clearly outweigh its weaknesses, this paper is suitable for publication.

The authors are strongly encouraged to carefully consider all of the reviewer feedback, including the rebuttal-related discussion, and to take meaningful steps to incorporate it into the final version of their paper.